# “That is an Awful Lot of Fruit and Veg to Be Eating”. Focus Group Study on Motivations for the Consumption of 5 a Day in British Young Men

**DOI:** 10.3390/nu11081893

**Published:** 2019-08-14

**Authors:** Stephanie Howard Wilsher, Andrew Fearne, Georgia Panagiotaki

**Affiliations:** 1Norwich Medical School, University of East Anglia, Norwich NR4 7TJ, UK; 2Norwich Business School, University of East Anglia, Norwich NR4 7TJ, UK

**Keywords:** young men, fruit, vegetables, psychology, visual research methods, conceptual framework

## Abstract

Young men do not consume enough fruit and vegetables, increasing their risk for future ill health. To understand what motivates their food choice, a novel conceptual framework that included key concepts from the theory of planned behavior and risk theory was developed. Thirty-four British men (18–24 years) took part in focus groups, where innovative visual qualitative methods provided insight into participants’ motivations for fruit and vegetable consumption. Based on information from food diaries, participants were described as high (4+ portions) or low (up to 3 portions) consumers. Interviews were coded thematically into concepts and characteristics of the conceptual framework. Young men were generally unaware of the UK government’s recommendation to consume 5 portions of fruit and vegetable a day and chronic health risks associated with low consumption. High consumers were motivated by perceived risk, perceived behavioral control, and health-conscious self-identity. They held internalized, holistic beliefs about diet and health, whereas low consumers’ beliefs were externalized, based on physical appearances. Low consumers were driven by social influences to consume cheap, easily available convenience foods. The conceptual framework differentiated levels of fruit and vegetable consumption between the two groups and provided new information about young men’s motivations for fruit and vegetable consumption.

## 1. Introduction

It is well documented that eating at least five portions of fruit and vegetables daily reduces the risk of all-cause mortality [1]. Despite the evidence for the health benefits of fruit and vegetables, average daily consumption is only three portions by much of the population across the world [2,3]. In England, around 50% of men consume fewer than three portions daily, with young men (18–24 years) eating the least [3,4]. This is of concern, as men are more likely than women to suffer chronic conditions in later life, such as coronary heart disease [5], and live around four years less [6]. Lifestyle differences between men and women may contribute to differences in health and mortality, as women are more likely to adopt healthier diets that include the consumption of healthy amounts of fruit and vegetables [7].

Existing research in this area has concentrated primarily on the determinants of fruit and vegetable consumption in *all adults* irrespective of age or gender ([8,9,10,11]. Studies that focus largely on young men, have concentrated on what influences healthy eating in general, creating a gap in our understanding of the barriers and motivators of fruit and vegetable consumption in this group of men [12,13,14]. Given the evidence that young men consume the least fruit and vegetables, which leaves them at risk of more chronic health conditions in later life, it is important to further explore the motivators and barriers of fruit and vegetables consumption in this particular group.

Barriers for healthy eating among young adults, include: male apathy towards diet; unhealthy diet of friends and family; low cost of unhealthy foods; lack of time to plan, shop, prepare and cook healthy foods; and lack of motivation to eat healthily. The desire for improved health, self-esteem and weight management were some of the motivators for healthy eating among young adults [12,13,14]. In a qualitative study addressing fruit and vegetable consumption in working men, Dumbrell and Mathai [15] reported that knowledge, time and effort, availability of fruit and vegetables, and women as gatekeepers to men’s diet, were both drivers and barriers to consumption among Australian men (18–40 years). Perceived barriers were low self-efficacy regarding cooking skills, negative attitudes to fruit and vegetables, and family preferences for unhealthy foods. Qualitative studies suggest that, although maintaining health is a motivator for healthy eating [12] and fruit and vegetable consumption in many adults [16], it does not drive the eating habits of all men and women [17] or men of all ages [15,16,18]. These inconsistent findings suggest that maintaining health as a motivator needs to be further explored in order to clarify its actual role in determining fruit and vegetable consumption in young men.

Although researchers have explored the barriers and motivators of young men’s health related behaviors in general, none of the available qualitative studies uses a valid theoretical framework that could provide a deeper understanding of what drives young men to consume the recommended five portions of fruit and vegetables daily [19]. The current research aims to advance our knowledge about young men’s fruit and vegetable consumption by using a conceptual framework broadly constructed around two major psychological theories – the theory of planned behavior (TPB) [20] and risk theory [21].

Most cited psychosocial theories and theoretical models (TPB) [20], social cognitive theory (SCT) [22], health belief model (HBM) [23], trans-theoretical model (TTM) [24], and health action process approach (HAPA) [25] — vary on coverage of the key concepts used to explain behavior. These include attitudes, self-efficacy, norms, risk and intention. For example, HBM covers attitude and risk, whereas HAPA, also includes intention and risk perception, however, risk is only measured on one construct. Fischhoff et al. [21] risk theory incorporates constructs that cover familiarity with and knowledge about risk, voluntariness of exposure and controllability over risk, benefits of risk and dread of consequences, immediacy of risk and effect of fatality.

It has been suggested that the TPB outperforms other models in predicting consumption of fruit and vegetables in the adult population [8]. One of its key concepts – perceived behavioral control (PBC) – has been consistently shown to predict behavior [26]. Internal beliefs (i.e., cooking skills and perceived time required to prepare a healthy meal), and external beliefs around availability and cost of fruit and vegetables, are also linked to adults’ consumption of fruit and vegetables [27].

Availability and cost are often designated as environmental factors [9], however, measurement of peoples’ perceptions make them contenders for perceived behavioral control. Studies have found that cost, availability and time often limit diets to unhealthier choices [13,28]. Attitudes (i.e., positive and negative evaluative beliefs) towards a behavior, and subjective norms (i.e., perceptions about what others believe a person should do) are weaker predictors of behavior [26].

The present study identified key concepts from the TPB reported to influence health-related behaviors, in order to assess whether these can also explain young men’s motivation to consume the recommended five portions of fruit and vegetables a day. A number of additional concepts were considered and adapted.

Subjective norm was replaced by social influences, to include the food environment, peers, social norms, and sources of information, all of which are known to influence fruit and vegetable consumption. For example, advertisements and the media have been identified as sources of information about diet by Belgian students [29].

Attitudinal measurement would also benefit from the inclusion of affective attitudes, that is how emotions impact on behavior [30], or liking of fruit and vegetables [31], and hedonistic attitudes, such as taste and satiety [8,32]. Self-identity, that is individually created to provide meaning and guide behavior, is an independent predictor of TPB for various behaviors [33]. For example, self-identification as a health-conscious consumer independently predicts lower consumption of animal fat [34], which inspired the term “health-conscious self-identity” (HCSI), describing people with salient health beliefs that influence their behavior.

The second theory often used in behavioral research is risk theory. Research on perception of risk suggests that although people understand risks to health, they do not perceive that their lifestyle choices might put them “at risk” [35,36,37]. The concept of trust has also been included in risk perception because it is integral to society and decision-making [38]. It covers general trust (i.e., understanding the need for policies) and skeptical trust (i.e., understanding how and why policies are made). People’s views are placed on a continuum between total rejection and complete acceptance of information about health and relevant policies. Being engaged and critically reviewing information is considered “healthy” trust [39], however, adults might legitimize their diet because of inconsistencies in health messages that lead to mistrust of the information source [17,35,40]. The key concepts of risk theory and trust offer a promising framework to assess whether risk to health motivates fruit and vegetable consumption in young men and were therefore included in the conceptual framework proposed by this study. 

Although historically TPB has proven successful at predicting fruit and vegetable consumption in adults, it lacks a measurement of risk to health, which can be achieved by the addition of risk theory constructs in the model. At the same time, risk theory has not been used to explain the consumption of fruit and vegetables - a gap that is addressed by the present study.

The proposed conceptual framework therefore combines key concepts from TPB and risk theory. Perceived behavioral control (self-efficacy and control characteristics), attitudes [8], with affective and hedonistic characteristics were added as these have been reported to influence health behaviors [30,31]. Subjective norm (TPB) was replaced by social influences, which included the characteristics of family and peer influences, social norms, and perceived environmental influences of food outlets and information. Health-conscious self-identity was included as it is a personalized concept, reported to independently predict health behaviors [33].

From risk theory, the concepts of knowledge about risk to health, benefit of fruit and vegetables to health, dread of illness, familiarity with illness, and immediacy of risk were employed. The concept of trust was added because of its link with risk perception and potential for behavior change [35]. Voluntariness and controllability were removed from the proposed framework as they broadly compare with TPB’s self-efficacy and control.

### Methodological Considerations

In addition to the limited research on fruit and vegetable consumption in young men, few studies have incorporated dietary measures to establish participants’ actual eating habits; only one study on older men included a measurement of fruit and vegetable consumption [18]. Most studies do not assess diet, which is a weakness as it is unclear whether the reported motivations really do drive behavior. People may evaluate that eating fruit and vegetables is good for health, however, other factors, such as cost, will lower motivation to eat 5 a day. It is, therefore, important to assess concepts of the theoretical framework from the perspective of fruit and vegetable consumption. To address this limitation, participants were asked to complete four food diaries or 24-h recalls. All dietary measurement are prone to errors; however, food diaries and 24-h recalls methods are quick to complete and four days is considered the optimal period for assessment [41] and using the two measures helped validate fruit and vegetable consumption [42].

This study used qualitative methods to explore young men’s motivations for fruit and vegetable consumption. Focus groups were conducted with three activities (i.e., selecting images from magazines, giving their views on health promotional material and suggesting ideas to promote fruit and vegetable consumption in young men), to provide variety for participants, prompt discussion, and gain different perspectives on diet, health and information. Visual images are social influences that are constructed to convey messages. Images may be interpreted as reality or as constructions of world complexity, allowing individuals to create their own personal meanings [43,44]. Much is written about how researchers can use visual images for research but little about using them to illicit perceptions from participants. Diet and health are commonly presented in magazines and health promotions, including 5 a day, making them ideal visual mediums to gain further understanding of the motivations of young men. A selection of these visual aids was used to explore personal experiences through perception that taps into emotional and cognitive processing. As far as the authors are aware, these innovative methods have not been used before to explore fruit and vegetable consumption, and health.

## 2. Materials and Methods

### 2.1. Participants

Inclusion criteria for this study were British men aged 18–24 years. Thirty-four participants (mean age 20.5 years) were recruited to eight focus groups. The number of participants in each group ranged between two to six. The sample’s demographic characteristics are presented in Table 1. None of the young men reported that they were vegetarians.

Young men were approached by the first author in public areas of a suburb of London, and a small rural market town in Norfolk, England, and asked if they would participate in a research study about diet and health. These areas were chosen to assess fruit and vegetable consumption in urban and rural settings. London is an excellent example of a large urban center whereas Norfolk is a relatively isolated rural area in the East of England, with a very different population makeup than London. The refusal rate was high, but unrecorded. Young men who agreed to take part in the research were invited to join a focus group in their area. Participants received £20 to cover any costs incurred. The University of Kent ethics committee in the UK approved the study.

To allow comparisons between levels of consumption, participants were segmented as high consumers (4+ portions) and low consumers (up to 3 portions). Using the cut-off of four or more portions was a compromise as only four men consumed 5 a day. Urban or rural setting was determined by address.

### 2.2. Procedure

Consenting participants were given details of the study in both verbal and written formats. They completed a consent form and were informed that they could withdraw from the study at any time. Participants provided demographic details and a food diary to capture food intake (breakfast, snack, lunch, snack, evening meal, snack and late-night snack) on a standard template, for the previous day. Three more food diaries or 24-h recalls were requested via telephone, text or email over the following month.

The focus groups were led by the first author and conducted in hired rooms in the two research areas. Each group lasted around 90 min and sessions were audio recorded. A semi-structured interview schedule explored factors influencing fruit and vegetable consumption within the conceptual framework. The interview schedule was adjusted after piloting with four 18–24-year-old men (Table 2).

At the start of each focus group, an icebreaker introduced participants who then named and gave reasons for their favorite foods, fruit and vegetables. Three activities were designed to maintain participant interest and provide alternative perceptions of health and diet. After discussions on fruit and vegetable consumption, participants were asked to select images they perceived as healthy or unhealthy from magazines available in England in April 2010: GQ, Men’s Health, Men’s Fitness, Men’s Running, Running, and an in-store supermarket magazine.

Following discussions on perceived health, participants were given a selection of health promotion materials available in England in 2010: Food Plate (Food Standards Agency -FSA), Change4life (National Health Service -NHS), 5 a day (NHS), cut down on salt and fat (British Heart Foundation), traffic light food labeling (FSA) and an information sheet on prostate cancer (Cancer Research UK), and asked to choose and comment on what they perceived as good and poor diet and health messages. This activity was designed to help us understand how young men perceive health and diet information.

Finally, participants were asked to imagine being employees of an advertising company and to create a message promoting 5 a day to young men (by kind permission of Dumbrell [15]). This activity generated ideas for promoting fruit and vegetable consumption to their peer group. At the end, participants were thanked, debriefed and given the opportunity to ask questions about the research.

### 2.3. Analysis

Fruit and vegetable consumption was assessed by counting the portions of fruit and vegetables recorded on the food diaries and 24-h recalls. A single piece of fruit, such as an apple, a glass of fruit juice, vegetables and pulses, such as baked beans, consumed at a meal also counted as one portion. Fruit or vegetables included in sandwiches were counted as half a portion. Consumption was averaged over four days.

Interview recordings were transcribed verbatim and pseudonyms used to safeguard confidentiality of participants. Personal data were stored securely and separately from the research data. NVivo 10 was used to thematically code into the characteristics and concepts of the conceptual framework. Transcripts concerning magazine picture choices and health promotion leaflets were analyzed in a similar way. These were particularly relevant to the ‘health-conscious self-identity (HCSI)’ and ‘social influences’ components of the proposed framework. Analysis was broadly based on thematic analysis outlined by Braun and Clarke [45], as it allows flexibility for analyzing data collected by different mediums. Data were coded line-by-line for each focus group, then iteratively collated into themes based on characteristics and concepts of the theoretical framework (see Figure 1). Any new themes that emerged were noted and reported. 

The first author coded all transcripts and each co-author independently coded twelve transcripts and consensus for any differences was reached through discussion.

## 3. Results

The findings of the thematic analyses were compared for differences between urban/rural and high/low fruit and vegetable consumption. The latter provided clear differences between young men. The findings are presented below for low consumers (<3 portions) and high consumers (4+ portions), under the concept and characteristic headings of the conceptual framework: PBC (self-efficacy, control belief), attitude (evaluative, affective, hedonic), social influences (family, peers, social norms, food environment, information), health-conscious self-identity (personal meaning) and risk (knowledge, benefit, dread, familiarity, immediacy, trust). Extracts are presented in italics as verbatim quotes selected to encapsulate each concept or characteristic. The location (London or Norfolk), focus group identifier and age in years follow each quote.

### 3.1. Perceived Behavioral Control: Self-Efficacy and Control

Control also showed differences between the two groups. High consumers felt they had good control of their diet and health: 

Myself, my own choice (London, C, 18 years).

Whereas, low consumers felt that others were in control:

If they (family) just cooked like fat foods all the time you are not going to be very healthy (Norfolk, G, 18 years).

Young men with high fruit and vegetable consumption expressed good self-efficacy regarding availability and cost of fruit and vegetables, the time and effort to buy, prepare and their good cooking skills enabled them to cook fruit and vegetables. Those with low consumption either could not or would not cook. For this group convenience foods were easier; fruit and vegetables were viewed as expensive and not readily available, and their preparation time-consuming (Table 3).

### 3.2. Attitudes: Evaluative, Affective and Hedonic

For both groups evaluative attitudes included belief that fruit and vegetables benefitted health, were nutritious and versatile, but were susceptible to damage and decay, and vegetables needed to be prepared and cooked. High consumers compared healthiness of fruit and vegetables with take-away meals:

(...) eating fruit and veg is a lot more healthy than going and getting a kebab (London, B, 18 years).

Low consumers considered vegetables in the context of a meal:

They are normally served with meat (London, D, 23 years).

High consumers expressed positive affective attitudes, such as enjoyment, satisfaction, alertness and emotional stability:

Moods, less snappy, more willing to do stuff to help, it totally changes everything (London, C, 22 years).

Low consumers expressed more negative affective attitudes, including strong feelings against vegetables:

Disgusting stuff, vegetables (Norfolk, F, 18 years).

Impatience and the need for immediate satisfaction from convenience foods was also discussed:

It’s like if I am hungry I want to eat it now (...) I am impatient and don’t like waiting (Norfolk, F, 20 years), and It hasn’t got any instant gratification of things to have lots of sugar (London, E, 19 years).

Low consumers also showed resistance to diet change:

(...) we have been told we need to eat and I think we go the other way (London, E, 23 years).

High consumers liked the taste and variety of flavors in fruit and vegetables, whereas low consumers felt that vegetables lacked flavor and that the taste of fruit was inconsistent. High consumers felt fruit and vegetables were filling, while low consumers felt convenience foods were better for satiety (Table 4).

### 3.3. Social Influences

#### 3.3.1. Family

Both groups expressed similar childhood experiences around food choice such as being bribed to eat fruit and vegetables:

Well you get told you need to eat it to get your pudding (London, E, 23 years).

High consumers recalled positive family influences:

(...) my mum because she cooks a meal and my brother as well because he became a chef and just started bringing a lot more healthy food (London, B, 18 years).

Only low consumers refused to eat fruit and vegetables:

I just refused it; I was quite picky (London, D, 24 years).

Low consumers thought behavior was learned from parents, but their mothers generally bought and cooked foods requested by their child:

My mum does cook stuff for me and she only usually cooks what I ask her to cook (London, B, 18 years).

Furthermore, this group expressed reliance on their mothers to provide meals with vegetables:

(...) I would only eat them (vegetables) if like mum were to cook them (Norfolk, F, 18 years).

Issues around learning to cook, and skills acquired during childhood were also raised. Most high consumers had experience of cooking at home and some had tuition at school or taught themselves. These young men expressed an openness to try recipes and new ideas:

I have got a few pointers off my mum and my nan, it’s kind of make it up as you go along and try things (London, C, 18 years).

Low consumers experienced little, if any tuition at home or school. This was particularly relevant to young men living in rural areas:

We didn’t get the chance; they didn’t do cookery at our school (Norfolk, F, 20 years).

#### 3.3.2. Peers

Low consumers generally embraced the poor diet of their group, and health was perceived through exercise or appearance with a drive to consume protein and build muscle. A friend introduced one low consumer to vegetables:

I have started to eat more leeks and cabbage now because I have friends that are Irish (London, D, 18 years).

Having a girlfriend improved fruit and vegetable consumption for a high consumer, but only raised awareness for a low consumer.

Life changes were not included in the conceptual framework but were raised as drivers and barriers to fruit and vegetable consumption by some participants. High consumers expressed personal motivation to trigger behavior change, such as to be able to work better or eat healthily. Low consumers viewed their social lives, hobbies, and living independently as factors that negatively influence their diet:

(...) my social life really and not having time to sit down and eat (Norfolk, F, 20 years).

I never ever think about my diet, I think about girls, motor cross, gym, cars (...) (Norfolk, F, 18 years).

I used to eat a lot more fruit and veg than I do now ‘cos it was easier back then (at family home) it was all prepared for me and now I have to actually cook for myself. It is just easier to stick something in the oven (London, D, 24years).

#### 3.3.3. Social Norms

High consumers did not appear affected by social norms (i.e., normative views held within society about certain behaviors). Low consumers were influenced by ‘stigma’ around fruit and vegetable consumption, and gendered norms of masculinity:

It is more of like a stigma attached to it (...) (Norfolk, F, 20 years)

(...) if it comes to like a proper meal then nah, it’s mum’s job (Norfolk, F, 18 years).

(...) not as many men probably cook so they eat what is given to them (Norfolk, G, 21 years).

#### 3.3.4. Food Environment

Both groups acknowledged that fruit and vegetables were prominently displayed in supermarkets, whereas convenience stores clearly displayed unhealthy foods. Some high consumers felt that educational establishments offered unhealthy food choices and provided little support for a healthy diet: They (colleges) mainly have plenty of pastries on offer and muffins and stuff like that and you think my god, no wonder everyone is getting fatter. High consumers also thought takeaway outlets should encourage healthy eating by including a greater variety of healthy food:

(...) give you more variety because I know if I get a salad from there (kebab house) I know it’s just going to be a lot of onion (Norfolk, H, 24 years).

#### 3.3.5. Information

Both groups were confused about what constituted a portion of fruit and vegetables, and disliked mixed messages around healthy and unhealthy foods. Both groups also liked the colorful pictorial representations of 5 a day. High consumers disliked the lack of serving suggestions and low consumers were deterred by the amount of fruit and vegetables:

That is an awful lot of fruit and veg to be eating so it deterred me straight away (Norfolk, A, 21 years).

Low consumers were aware of the controversy surrounding the recommended number of portions and understood that the recommended 5 a day was low compared with other countries.

Two low consumers had seen the more recent Change 4 Life campaign on television. Low consumers were drawn to the Change 4 Life message by its informal and bright presentation but were deterred by all the writing and family focus.

Both groups liked the Food Standards Association (FSA) plate because it is applicable to daily life and motivating:

I think it is very quick and easy to see like everyone can relate to it (...) (London, D, 24 years, low consumer).

Low consumers felt some of the pictures, such as those of raw fish, were repellent and lacked guidance around serving suggestions and health risks:

It’s not even scary; I don’t even think it’s that well done (Norfolk, E, 19 years).

Both groups thought the Healthy Heart messages for reducing salt and fat were poorly presented, ambiguous and lacked information on risk to health:

(...) to be honest he doesn’t exactly look the healthiest of guys (...) who’s trying to say ‘hang on I don’t do it myself’ (Norfolk, E, 20 years, low consumer).

Finally, participants felt that diet and health promotions should be designed around their interests, such as sex, exercise and sports. Fruit and vegetables could be promoted at the point of purchase, in fitness clubs, in men’s magazines and in free media spaces. Promotions should be developed and presented appropriately to provide real examples of male health and fitness.

### 3.4. Health Conscious Self-Identity

Choices of healthy and unhealthy images provided further insight into how participants viewed diet and health. High consumers chose an image of a group of young people relaxing together by a swimming pool, which they felt represented social health as well as physical and mental well-being. For low consumers, the same picture was perceived as an advertisement for designer clothing. Another image represented a healthy diet (vegetables, nuts, pasta) for high consumers. Low consumers chose this picture because it highlighted the impact of diet on mental health.

High consumers presented an internalized, holistic view of health, including social health, where successful relationships are enjoyed. They viewed health as a responsibility, something that should be nurtured:

It’s a responsibility (...) and its lifestyle and the way you perceive and live life (London, B, 18 years).

For low consumers, good health was recognized as important for daily life, but it was an aspiration rather than a priority:

A single male, it’s (health) not really on the top of your mind too much, there are always other things to be doing (London, D, 24 years).

External attributes such looking good, being fit and muscular, were key indicators of health for low consumers:

Just go on how ripped (muscled) someone is (Norfolk, F, 18 years).

These young men consumed energy/recovery drinks and went to the gym to keep fit.

### 3.5. Risk

Young men were familiar with chronic health conditions (diabetes and heart disease) in their family but many were unaware that fruit and vegetable consumption could lower the risk of developing these conditions. High consumers equated health with normal weight and dreaded becoming ill. High consumers controlled their diet and consumed fruit and vegetables for general health. These young men wanted long-term health and showed critical trust in health promotions and dietary advice.

Low consumers reported chronic illnesses within their family more often than high consumers, and some were being monitored for early signs of chronic conditions. Low consumers were aware of the relationship between fat and artery health and exercised for protection. These young men hoped that they would not become ill but did not think about their future health. Most low consumers distrusted health promotions and felt young people would rebel against government promotions (Table 5).

## 4. Discussion

This study explored motivations for fruit and vegetable consumption in 18–24-year-old British men. The study assessed a novel conceptual framework that combined key concepts of two established theories - the theory of planned behavior and risk theory. Transcripts from a series of focus groups with young men were coded, analyzed and separated into high and low consumers of fruit and vegetables. Overall, the conceptual model highlighted similarities and differences in motivations of the high and low fruit and vegetable consumers. The addition of risk, HCSI and social influences provided information on further motivations not covered in the TPB. However, as discussed below, it is likely that there is an interaction between these concepts. For example, perceived risk to health might influence attitudes and perceived behavioral control. Perceived risk to health could also be swayed by social influences, which is much wider ranging compared to subjective norm found in TPB.

Concepts from the proposed framework such as perceived behavioral control (PBC), hedonic and affective attitude, health-conscious self-identity (HCSI), and risk perception motivated high consumers but not low consumers. Social influences motivated low consumers, whereas evaluative attitudes, knowledge, and health information were similar for both groups. Both groups presented the same evaluative attitudes around fruit and vegetable preparation and their nutritional qualities. The young men in the study had a basic understanding of general health, but none appreciated the connection (knowledge) between fruit and vegetable consumption and chronic health conditions. They also seemed unaware of health information and the 5-a-day campaign (social influences).

Consistent with earlier research [10,18,46,47], the findings indicated that motivation to consume fruit and vegetables differed between the two groups. High consumers showed high self-efficacy and control over their diet and health. These characteristics were missing in low consumers who lacked cooking skills and lacked time and effort for preparing healthy meals. Perceived cost and poor availability of fruit and vegetables were also barriers. Perceived behavioral control is an effective predictor of fruit and vegetable consumption [8,10,26], but it assumes behavior is volitional. This study highlights that PBC might be mediated by meal providers and stresses the importance of social influences that go beyond subjective norms in TPB [8].

Family factors positively influenced high consumers but negatively affected low consumers’ choices - a finding that has previously been reported in studies with children [48,49,50]. Although living alone was not a factor included in the conceptual framework, it appeared to negatively affect some young men’s choices - a finding consistent with research in widowed older men [51]. Social norms, especially perceptions of masculinity and conformity to gendered roles are known to influence men [15,52]. In the present study, this influence, derived from family, peers and information sources was evident in low consumers. Gendering and medicalization of diet and health proliferates in social influences [53,54,55]. Many young men felt there was lack of support for a healthy diet within the food environment, especially educational establishments, which has also been reported by university students [16].

High consumers expressed positive hedonic and affective attitudes around fruit and vegetable consumption. Low consumers showed negative hedonic attitudes and emotions and often referred to poor mental health, which might be due to poor diet [41,56,57]. Low consumers combined satiety (hedonic) with instant gratification (affective attitude), which might indicate poor emotional health and planning skills. This has also been identified in women with depression [58]. Instant gratification is new among adults, but has been identified in children and adolescents [50]. The taste of fruit and vegetables was a driver for high consumers and barrier for low consumers, suggesting that it might be a strong predictor of attitude [8]. This research supports the inclusion of affective and hedonistic attitudes to improve attitudinal prediction of TPB [30,31]. Research also suggests that social norms could influence dietary tastes [59] and health-conscious self-identity [15,60].

Health-conscious self-identity is important [33,34], and the findings support its inclusion in the conceptual framework, as novel differences between the two groups of consumers were identified. High consumers selected magazine pictures on interpretive meaning, suggesting that they perceived health holistically: this included the physical, mental and social benefits of healthy eating and fruit/vegetable consumption. Low consumers often focused on external markers of health, such as muscularity. Their choice of pictures was based on literal interpretation and for reference to mental health. HCSI may serve as a proxy for risk theory, however, characteristics from risk theory provided insight into how young men perceived risk and how it impacted on fruit and vegetable consumption.

The addition of ‘trust’ (risk) in the proposed conceptual framework provided insight into young men’s perceptions of health information and the degree to which these influence dietary choices. The high degree of mistrust towards sources of health information identified in this study indicates the need to adopt different approaches to engage young men with health messages and improve their dietary choices. High consumers perceived unhealthy eating as a risk to general health, considered their long-term health and adopted healthy lifestyle behaviors because they dreaded becoming ill. Despite suffering from more chronic conditions, low consumers hoped for good health, regulated by exercise. In line with previous research [18,35,61], low consumers did not consider dietary change for health benefits. Hartman et al. [16] also reported that university students did not consider the importance of their health unless prompted. Indeed the “live for today” ethos of low consumers suggests optimistic bias toward health [37], which might partly explain the distrust and skepticism of health promotions and their sources [39].

Using health promotions and magazines provided self-concepts of diet and health risk based on how the information about healthy eating was perceived by the young men [43,44]. High and low consumers interpreted pictures differently which gave further texture to the findings, especially around health-conscious self-identity, and highlighted how these social influences motivate young men’s fruit and vegetable consumption. The use of suitable magazines provided a standardized research approach and gave participants the opportunity to give their own interpretations on what the visual images meant to them in terms of diet and health. It is novel to use visual research methods in this way and further research is needed to provide better methodological understanding. 

The sample size was large and consisted of young men in various employment situations. Four food diaries and/or 24-h recalls were used to measure fruit and vegetable consumption. The number of portions the young men consumed determined the motivations of high and low consumers. This is important to fully understand the motivations of this demographic group to develop and target interventions.

All participants were white from two areas of the UK, which may limit transferability to other ethnic groups and cultural contexts. Volunteering to take part in focus groups limited the sample to motivated young men who may not be representative of those who did not volunteer.

Any self-reported dietary measure is a proxy for observed intake and using it highlighted the problem that participants may not report fruit and vegetable consumption accurately, unless prompted, thus potentially resulting in under-reporting. Nevertheless, use of two measures, food diary and/or 24-h recall, enabled data triangulation and improved reliability of fruit and vegetable consumption over four days. Finally, since this study was completed, new social media, such as telephone apps, have become mainstream. Future research could assess the social influence of these media outlets on young men’s consumption of fruit and vegetables. 

Further research should explore perceived risk to health and HCSI. Likewise, research is needed to explore new characteristics, life changes, instant gratification and trust identified herein. In addition, including the study of other health-related behaviors such as smoking or alcohol consumption could provide researchers with a more detailed exploration of the complex picture around young men’s motivation to consume fruit and vegetables. Given the paucity of research on young adults, this research should be replicated in other countries around the world to assess personal and cultural differences in diet and health perception. The findings can inform the design of tailored interventions for the improvement of fruit and vegetable consumption in young men. Lack of cooking skills in this population should be addressed, ideally combined with nutrition, health education and physical exercise. In addition, health promotions for fruit and vegetables should be framed to capture young men’s interests and displayed in places they frequent.

## 5. Conclusions

By using dietary measures, our conceptual framework revealed new information and different motivations between young men with high and low fruit and vegetable consumption. The use of novel visual methods to explore health and diet provided further understanding of these motivations. In particular, HCSI and whether this was internalized motivated behavior. Affective and hedonic motivations were different between the groups. Many social influences effect the motivations of low consumers. Young men were generally unaware of the UK government’s recommendation to consume 5 portions of fruit and vegetable a day and chronic health risks associated with low consumption. However, mistrust in health promotions and media sources provide major challenges for increasing fruit and vegetable consumption. Finally, health promoters and policy makers should use these findings to model and deliver holistic interventions, focused on the needs of young men.

## Figures and Tables

**Figure 1 nutrients-11-01893-f001:**
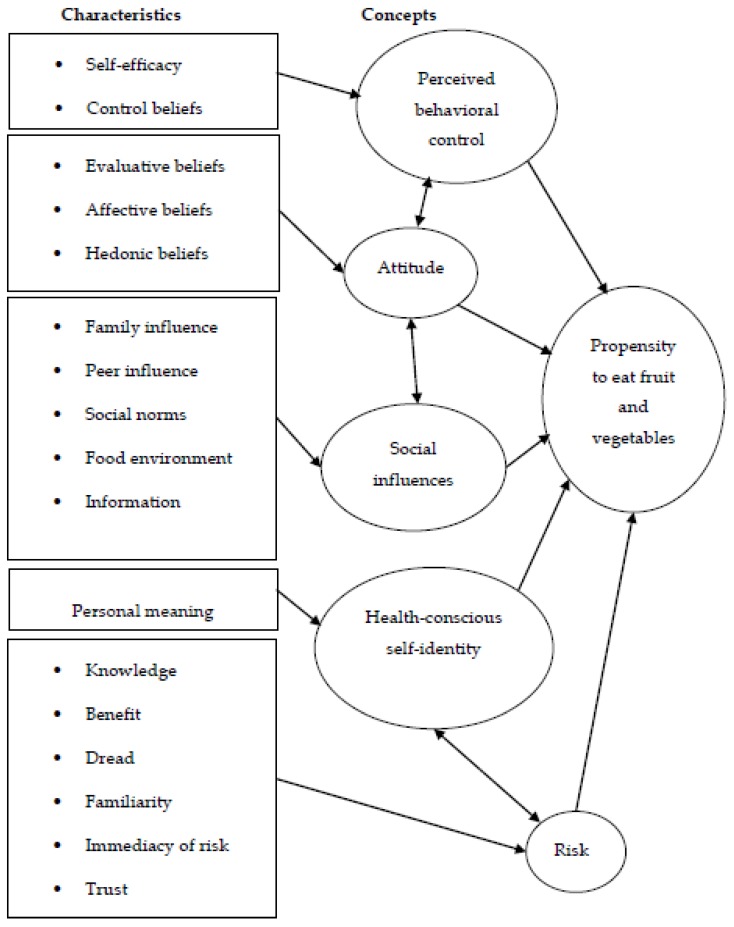
Proposed conceptual framework for the exploration of fruit and vegetable consumption in young men.

**Table 1 nutrients-11-01893-t001:** Demographic characteristics of participants in the focus groups.

Characteristic	Level	Number of Participants
Age	18	11
	19	4
	20	2
	21	6
	22	2
	23	3
	24	6
Fruit and vegetable consumption	High	10
	Low	24
Area	Urban	22
	Rural	12
Education	Graduate	3
	6th form/further training	22
	High school	11
Work status	Skilled workers	21
	Semi-skilled workers	7
	Manual workers	2
	Student	3
	Unemployed	3

**Table 2 nutrients-11-01893-t002:** Interview guide for the focus groups.

Concepts in the Conceptual Framework	Example Interview Questions
Perceived behavioral control	What helps/stops you eating fruit/vegetables?
Attitude	What are the good things about fruit/vegetables? What is the trouble with fruit/vegetables? Why do you think young men eat less fruit/vegetables than other groups?
Social influences	Who influences your fruit and vegetable consumption? Who do you think looks after your health? What would help young men to think more about their health? Where do you find information on diet and health?
Health conscious self-identity	How would you describe your health? What does your health mean to you?
Risk to health	What are the benefits of eating fruit/vegetables? Would you say your diet affects your health? Have you experienced any health issues?

**Table 3 nutrients-11-01893-t003:** Self-efficacy for fruit and vegetable consumption.

Characteristic	Level of Fruit and Vegetable Consumption	Verbatim Quotes
Cooking skills	High	I have to plan mine a lot, plan my meals for the next day (London, C, 24 years). I just make all the sauces more or less the same but like change them slightly so, it is easy, all in one pan (London, C, 22 years).
	Low	(...) now I have to actually cook for myself It is just easier to stick something in the oven (London, D, 24 years). (...) rubbish at cooking (London, D, 24 years).
Time and effort	High	Fruit, it’s there and I can eat it while I am driving, it’s convenient (London, D, 19 years).
	Low	(...) takes a bit more time (preparing vegetables) than just going off and getting a chocolate bar (London, B, 18 years).
Availability of fruit and vegetables	High	It has always been there (...) Go grab an apple (London, C, 21 years). You can go down most high streets and there is bound to be a fruit and veg store (London, C, 21 years)
	Low	There is never really that much fruit and veg in our house Norfolk, F, 20 years). It’s finding the nice ones in the supermarket as they usually get damaged when being shipped (London, C, 24 years).
Perceived cost of fruit and vegetables	High	I can go and buy apples for the same price (as a chocolate bar) and it will last me from the morning all the way till the evening (London, B, 19 years).
	Low	You go to a shop and look at the prices sometimes and think its 40p for an apple, 40p for a chocolate bar, chocolate bar tastes nicer though (London, D, 20 years).

**Table 4 nutrients-11-01893-t004:** Hedonic attitudes of young men with high and low fruit and vegetable consumption.

Hedonic Attitude	Level of Fruit and Vegetable Consumption	Verbatim Quotes
Taste	High	They (vegetables) come in all different flavors (London, B, 24 years).
	Low	(...) some of them (vegetables) are pretty bland; I am not a huge fan of them really. I wouldn’t base a meal around them (Norfolk, H, 18 years).
Texture	High	Some people like it like a bit crunchy and some people like it soft (London, B, 18 years)
	Low	I don’t like the texture of most of it (fruit) so I don’t eat it (Norfolk, A, 18 years)
Satiety	High	A packet of apples (...) will fill me up for the whole day (London, B, 19 years).
	Low	Something like that (packet of crisps) fills me up more than say like an apple would (...) (Norfolk, A, 21 years)

**Table 5 nutrients-11-01893-t005:** Risk theory characteristics by high and low consumers of fruit and vegetables.

Concept	Level of Fruit and Vegetable Consumption	Verbatim Quotes
Knowledge of risk	High	If you have a bad diet, then you are going to be unhealthy (London, C, 24 years)
	Low	I am (not) helping my way to diabetic really because I do get quite a lot of exercise (Norfolk, F, 20 years)
Benefit of eating fruit and vegetables	High	(...) being that large isn’t good for your heart and health so vegetables is the way (London, C, 22 years)
	Low	I don‘t know really, I’m not educated enough (London, E, 23 years)
Dread of illness	High	We don’t want to be all fat and having heart attacks at 40 (...) so we stick to fruit (London, C, 18 years)
	Low	I am hoping I haven’t got it (high cholesterol) (Norfolk, H, 18 years)
Familiarity with disease	High	(...) my auntie has diabetes, cancer (...) (Norfolk, G, 18 years)
	Low	High cholesterol (...) Granddad and my mum (...) me and my brother had to be tested for it (Norfolk, F, 18 years)
Immediacy of risk	High	(...) I do it now you are going to be able to live a really long life (Norfolk, H, 24 years)
	Low	Live for today (Norfolk, G, 21 years)
Trust in health promotions	High	I trust Sally [his partner and nutritionist], I‘d probably pay more attention than a sign in a supermarket. I‘ll probably look into it and find out really ( Norfolk, G, 18 years)
	Low	(...) people will think it’s the government nagging other people to lose weight (Norfolk, G, 21 years)

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
