# Peer review of "“That is an Awful Lot of Fruit and Veg to Be Eating”. Focus Group Study on Motivations for the Consumption of 5 a Day in British Young Men"

_nutrients, 2019, doi:10.3390/nu11081893_

Round 1

Reviewer 1 Report

An interesting and well written paper that describes the barriers and influences of fruit and vegetable consumption in a group of young men - which will contribute towards health promtion policy around fruit and vegetable consumption.

However, there are many typos throughout the text and this manuscript needs to be sufficiently edited. For example

page 1 line 31 - "porions" should be "portions"

page 2 line 45  - it "is" important

It also hard to determine the participants quotes through out the text - some are italicized, some are not.  rarely are they indented as separate paragraphs, and there is not much information about who the speaker is ie age etc.

page 5 line 169 - What is the specific ethics committee and protocol number.

Author Response

Dear Review 1. Thank you very much for your positive feedback on our manuscript. We have addressed your comments below and in the manuscript as detailed: 

An interesting and well written paper that describes the barriers and influences of fruit and vegetable consumption in a group of young men - which will contribute towards health promtion policy around fruit and vegetable consumption.

1. However, there are many typos throughout the text and this manuscript needs to be sufficiently edited. For example

page 1 line 31 - "porions" should be "portions"

page 2 line 45  - it "is" important

We have checked other spellings and those still highlighted in spell checker are because UK English has been used throughout, rather than US English. The study was conducted in the UK, but we will change these spellings if required by the editor. 

Page 1, line 31 - this this typo has been corrected.  

Page 2, line 45 - "is" has been added. 

2. It also hard to determine the participants quotes through out the text - some are italicized, some are not.  rarely are they indented as separate paragraphs, and there is not much information about who the speaker is ie age etc.

All quotes are now italicised and separated into their own paragraphs to improve the reading. We have added the location, focus group identifier and age for each quote. See page 7, line 240: The location (London or Norfolk), focus group letter and age following each quote. We used UK spelling throughout, but we can change it to US English if required.

3. page 5 line 169 - What is the specific ethics committee and protocol number.

Page 5, line 170 - The University of Kent ethics committee has been named. At the time a protocol was not required for this study. 

Reviewer 2 Report

In the manuscript That is an awful lot of fruit and veg to be eating”. Focus group study on motivations for the consumption of 5 a day in young men, authors provide valued information about the motivations for the fruit and vegetables consumption in the UK. My comments and questions are listed below:

1.       In the title, the quote “That is an awful lot of fruit and veg to be eating” is an example of the attitude from the low consumption group, and I think it is not necessary to be emphasized in the title, also it makes the title too long. In that way the project name “5 a day” can be replaced with fruit and vegetables consumption in the young men in the UK.

2.       As for the research design, my main concern is the participants’ inclusion criteria. In the part of 2.1, page 5 line 161, authors only indicated that British young men aged 18-24 years are included. Did the authors consider other criteria such as education level, occupation, vegetarian, drinking/smoking?

3.       The participants are from two areas in the UK, suburb of London and rural market town in Norfolk. Could the authors explain why they choose these two area for target young men.

4.       In the current manuscript, the motivations for the fruit and vegetables consumption are discussed. Did the authors have any data or findings regarding the difference between the motivations for fruit and vegetables?

Author Response

Dear reviewer 2. Thank you very much for your positive feedback on our manuscript. We have addressed your comments below and in the manuscript as detailed: 

1.In the title, the quote “That is an awful lot of fruit and veg to be eating” is an example of the attitude from the low consumption group, and I think it is not necessary to be emphasized in the title, also it makes the title too long. In that way the project name “5 a day” can be replaced with fruit and vegetables consumption in the young men in the UK. 

We have modified the title in accordance with your recommendations:  

Focus group study on motivations for the consumption of fruit and vegetable consumption in young men in the UK.

2.As for the research design, my main concern is the participants’ inclusion criteria. In the part of 2.1, page 5 line 161, authors only indicated that British young men aged 18-24 years are included. Did the authors consider other criteria such as education level, occupation, vegetarian, drinking/smoking? 

Inclusion criteria was age 18-24 years, however, we included educational status, occupation and vegetarianism on the demographics questionnaire. Education status and occupation has been added to Table 1 on page 5 and on line 164 we added: None of the young men reported that they were vegetarians.

In hindsight, it would have been helpful to add questions about smoking/drinking as research suggests that participants often pursue several unhealthy behaviours. We have added a new sentence on page 12 lines 495-497 as a limitation and suggestion for future research. 

3.The participants are from two areas in the UK, suburb of London and rural market town in Norfolk. Could the authors explain why they choose these two areas for target young men.

The reason for choosing two areas - urban and rural was justified in line 166. Choosing participants from a highly urbanised area – such as London – and comparing them with participants from a rural area might have shown different findings. 

We have added the following sentence on page 5, line 169: London is an excellent example of a large urban centre whereas Norfolk is a relatively isolated rural area in the East of England, with a very different population makeup than London. 

Upon analysis, little differences between areas was noted, except that young men in the rural area had little taught cookery at school, compared to those in the urban area - this was reported on page 9, line 302.

4. In the current manuscript, the motivations for the fruit and vegetables consumption are discussed. Did the authors have any data or findings regarding the difference between the motivations for fruit and vegetables?

There were few differences in motivations for fruit or vegetable consumption; therefore, these data were presented for both fruit and vegetables. 
